Published at Building Trust Workshop at ICLR 2025

# EVALUATING TEXT HUMANLIKENESS VIA SELF-SIMILARITY EXPONENT

**Ilya Pershin**
Research Center of the Artificial Intelligence Institute
Innopolis University
Innopolis, Russia
`i.pershin@innopolis.ru`

## ABSTRACT

Evaluating text generation quality in large language models (LLMs) is critical for their deployment. We investigate the self-similarity exponent S, a fractal-based metric, as a metric for quantifying "humanlikeness." Using texts from the public available dataset and Qwen models (with/without instruction tuning), we find human-written texts exhibit S = 0.57, while non-instruct models show higher values, and instruct-tuned models approach human-like patterns. Larger models improve quality but benefit more with instruction tuning. Our findings suggest S as an effective metric for assessing LLM performance.

## 1 INTRODUCTION

The remarkable capabilities of large language models (LLMs) in generating human-like text have sparked significant interest in understanding the underlying mechanisms that enable such performance. One promising perspective lies in the study of the fractal structure of language, which offers insights into the self-similarity and long-range dependence (LRD) inherent in natural language. Recent research has shown that these properties are not only fundamental to the nature of language but also play a critical role in the success of LLMs Alabdulmohsin et al. (2024).

Self-similarity, a concept introduced by Kolmogorov (1940) and popularized by Mandelbrot (1983), refers to the property of an object or process being statistically invariant across scales. In the context of language, this means that patterns observed at smaller granularities (e.g., words or clauses) mirror those at larger scales (e.g., paragraphs or documents).

Despite the growing recognition of the importance of fractal properties in language, most traditional evaluation metrics for LLMs, such as perplexity-based bits-per-byte (BPB), fail to capture these deeper structural characteristics. To address this gap, recent studies have proposed using fractal parameters like the self-similarity exponent S, Hurst parameter H, and Joseph exponent J as alternative or complementary metrics for assessing model quality Alabdulmohsin et al. (2024). These parameters not only correlate strongly with downstream performance but also offer insights into the intrinsic complexity of generated text.

In this study, we investigate the use of the self-similarity exponent S as a metric for evaluating the humanlikeness of machine-generated text. Specifically, we compare texts produced by different versions of the Qwen model Bai et al. (2023), including both instruction-tuned and non-instruction-tuned variants, against human-written texts from the MAGE public available dataset Li et al. (2024). Our findings demonstrate that instruction tuning significantly improves the alignment of machine-generated text with human-authored content, as measured by S.

## 2 METHODOLOGY

The self-similarity exponent S is a measure used to quantify the degree of self-similarity in a time series or sequence. To compute the self-similarity exponent S, we followed the approach Alabdulmohsin et al. (2024). We used a large language model (LLM) to calculate the probability of the next token $w_t$ conditioned on its preceding context $w_{[t-1]} = (w_0, w_1, ..., w_{t-1})$. The corresponding

Table 1: Self-similarity exponent S for texts generated by different variants of the Qwen model. For human-written texts, we observed a self-similarity exponent $S = 0.5753 \pm 0.09$.

|  | **Qwen 1.5B** | **Qwen 3B** |
| --- | --- | --- |
| W/ instruction tuning | $0.6654 \pm .0228$ | $0.6462 \pm .0179$ |
| W/o instruction tuning | $1.0412 \pm .0368$ | $0.7273 \pm .0551$ |

number of bits for each token is then calculated using the formula:

$$z_t = -\log p(w_t | w_{[t-1]}).$$

The self-similarity exponent S quantifies the degree of self-similarity in a time series. Formally, a process $(X_t)_{t\in\mathbb{N}}$ is self-similar if it satisfies the following property:

$$(X_{\tau t})_{t\in\mathbb{N}} \overset{d}{=} (\tau^S X_t)_{t\in\mathbb{N}}$$

Here, $\tau$ is the granularity level, and S is the self-similarity exponent. To estimate S, we followed the procedure described below:

1. We constructed the increment process $(x_t)_{t\in\mathbb{N}}$ by normalizing the bit sequence $z_t$ to have zero mean and unit variance.

2. The integral process $(X_t)_{t\in\mathbb{N}}$ was computed as the cumulative sum of the increment process.

3. For a fixed small value $\epsilon << 1$ we calculated the probability mass function $p_\epsilon(\tau)$ of the event $|X_{t+\tau} - X_t| \leq \epsilon$.

4. Finally, we estimated S by fitting a power law relation to $p_\epsilon(\tau) \sim \tau^{-S}$.

The value of S was obtained by performing a linear regression on the logarithmic scale of $p_\epsilon(\tau)$ versus $\tau$. In our experiments, we set $\epsilon = 5 \times 10^{-3}$, as this choice has been shown to yield robust results Alabdulmohsin et al. (2024).

To evaluate the self-similarity exponent, we used the MAGE dataset Li et al. (2024), which consists of human-written texts. From the dataset, we selected 1000 samples for further generation. For machine-generated texts, we employed the Qwen models Bai et al. (2023) with varying parameter counts (1.5B and 3B) and instruction-tuned variants. Each model generated comments based on the provided prompt:

> "Write your comment based on the text below as if you were a member of an online forum. The comment should be approximately 400 words.
> {TEXT}",

where {TEXT} represents human-written content from the MAGE dataset.

We cropped both human and generated texts to 1024 tokens to ensure identical conditions. We used the Qwen 7B model to compute the self-similarity exponent for human and generated texts.

## 3 RESULTS

We evaluated the self-similarity exponent S for texts generated by different versions of the Qwen model, including both instruction-tuned and non-instruction-tuned variants. The results are summarized in Table 1.

For human-written texts, we observed a self-similarity exponent $S = 0.5753 \pm 0.09$, which aligns with prior study Alabdulmohsin et al. (2024). For machine-generated texts, significant differences were found depending on the model architecture and whether instruction tuning was applied. Non-instruction-tuned models exhibited higher S values, indicating greater divergence from human-like text patterns. For example, Qwen 1.5B and Qwen 3B without instruction tuning yielded

$S = 1.0412 \pm 0.0368$ and $S = 0.7273 \pm 0.0551$, respectively. Instruction-tuned models produced outputs with significantly lower S values, approaching those of human-authored texts. Qwen 1.5B-Instruct and Qwen 3B-Instruct achieved $S = 0.6654 \pm 0.0228$ and $S = 0.6462 \pm 0.0179$, respectively.

## 4 CONCLUSION AND DISCUSSION

This study investigated the use of the self-similarity exponent S as a metric for evaluating the human-likeness of machine-generated text produced by different versions of the Qwen model. Our findings demonstrate the importance of instruction tuning in aligning LLM outputs with human expectations, even when controlling for model size. While larger models generally improve text quality, the benefits diminish without proper alignment through instruction tuning.

Our findings contribute to building trust in large language models (LLMs) by introducing a new lens through which to evaluate their outputs. The ability of S to distinguish between human-written and machine-generated texts suggests its potential as a practical tool for advancing the development of more reliable and human-like models.

While this study highlights the utility of S as an evaluation metric, several avenues remain open for further exploration:

- Investigating the behavior of S in domain-specific contexts, such as scientific writing, legal documents, or creative fiction, could provide insights into its applicability in specialized settings.
- While this study focused on the Qwen family of models, future work should investigate the behavior of S across different families of LLMs, such as PaLM [6], GPT [49], and LLaMA.

In conclusion, our work contributes to the growing body of literature on the fractal structure of language and its implications for LLM evaluation. By leveraging S as a metric, we aim to bridge the gap between theoretical insights and practical applications, ultimately fostering the development of more trustworthy and human-like language models.

ACKNOWLEDGMENTS

All authors were supported by the Research Center of the Artificial Intelligence Institute of Innopolis University.

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
