# OpenReview forum: "Evaluating Text Humanlikeness via Self-Similarity Exponent"
_ICLR.cc/2025/Workshop/BuildingTrust — BuildingTrust_

### Official Review · Reviewer_sSwu · 2025-02-28
**Good empirical validation, but the experimental setup is limited**

**Rating:** 3
**Confidence:** 2

**Review:**

**Summary**

This paper suggests using the self-similarity exponent S to assess the humanlikeness of text generated by large language models. Experiments compare human-written texts from the MAGE dataset with outputs from different variants of the Qwen model. The findings show that human texts exhibit S ≈ 0.57, while non-instruction-tuned models yield higher S values. Notably, instruction tuning significantly lowers S, bringing machine-generated texts closer to human-like patterns.

**Strengths**
* **Empirical Validation**: By comparing different model variants (instruction-tuned vs. non-instruction-tuned) and relating S to human-written texts, the study provides compelling evidence that instruction tuning can improve humanlikeness.

**Weaknesses**
* **Limited Model Scope**: The experiments are confined to the Qwen family of models. Evaluating S across a broader range of language models would strengthen the claim of its general applicability.
* **Influence of generation parameters**: I believe that the humanlikeness of model generation depends on the sampling parameters (temperature, top p, top k, number of beams, etc.). The article does not specify which parameters were used in the study. It would also be interesting to look at the metric values with other values.
* **Comparative Analysis**: The paper could benefit from a more extensive comparison with other evaluation metrics to highlight the unique advantages and potential limitations of using S.

**Questions**
* Could you tell us about calculating self-similarity exponent S using some toy example so that the algorithm is better understood?

---

### Official Review · Reviewer_NSFe · 2025-03-03
**While the approach of this paper (using the self-similariy exponent as a metric  for evaluating the human-likeness of machine-generated text)  is novel and in scope of the workshop, more invest5igations into its applicability and usefulness is necessary.**

**Rating:** 6
**Confidence:** 4

**Review:**

This paper presents an interesting contribution to evaluating the human-likeness of LLM-generated text using the self-similarity exponent $S$, a fractal-based metric. The results effectively demonstrate that instruction tuning significantly improves alignment with human text patterns, leveraging $S$. However, the study has several limitations, as the real world interpretability and practical usability remain unclear of this metric. How does this metric compare to more traditional evaluation methods such as perplexity or human annotation in assessing trustworthiness? Additionally, investigating how $S$ correlates with human perceptions of text quality could help establish its real-world relevance.

---

### Decision · Program_Chairs · 2025-03-04

Accept